# Delay in Flowering Time in *Arabidopsis thaliana* Col-0 Under Water Deficit and in the *ddc* Triple Methylation Knockout Mutant Is Correlated with Shared Overexpression of *BBX16* and *BBX17*

**DOI:** 10.3390/ijms26178360

**Published:** 2025-08-28

**Authors:** Emil Vatov, Tsanko Gechev

**Affiliations:** 1Center of Plant Systems Biology and Biotechnology, 4023 Plovdiv, Bulgaria; gechev@cpsbb.eu; 2Department of Molecular Biology, Plovdiv University, 4000 Plovdiv, Bulgaria

**Keywords:** abiotic stress, *Arabidopsis thaliana*, cytosine methylation, drought, epigenetics, flowering, water deficit

## Abstract

This study addresses the role of cytosine methylation in the fine-tuning of flowering time under water deficit in *Arabidopsis thaliana.* A *drm1 drm2 cmt3 (ddc)* triple methylation mutant was used together with the Col-0 wild type. The plants were grown under long-day conditions with water deficit induced by cessation of watering starting 12 days after seeding. Col-0 showed a 1-day delay in flowering as a result of the treatment. In contrast, *ddc* showed a 2-day delay regardless of the experimental conditions. We found that the two b-box domain proteins, BBX16/COL7 and BBX17/COL8, became overexpressed in the *ddc* background and in Col-0 under water deficit 24 days after seeding. Additionally, the NF-YA2 transcription factor became correspondingly down-regulated. Our results support a model where BBX16/COL7 and BBX17/COL8 interact with CONSTANS to delay the induction of *FT* under long-day conditions. *NF-YA2*, which is also recognized as a promoter of *FT* expression, with its down-regulation causes additional delay of *FT*-induced flowering. The plants overcome the BBX/NF-YA inhibition easily, resulting in a relatively small delay in flowering. The expression patterns of the three genes suggest the involvement of cytosine methylation in their regulation; however, no differential methylation could be found in *cis* that can explain these effects. The results therefore suggest a *trans* acting mechanism. Considering that the activities of *BBX16/COL7* and *BBX17/COL8* in different physiological conditions are not elucidated, this paper provides a background for future experiments targeting the role of these genes in the fine-tuning of flowering time in *A. thaliana*.

## 1. Introduction

Stress-induced disturbances in flowering time are of increasing importance, not only due to the lack of irrigation water in many parts of the world, but also due to global climate changes leading to sporadic hot and dry spells. Better understanding of the molecular mechanisms underlying flowering time under drought can help breeders and farmers utilize varieties better suited for their conditions. For example, a region experiencing long dry spells may benefit from varieties that induce earlier flowering and avoid the stress altogether. On the other hand, areas with relatively short and sporadic dry spells may benefit from varieties that can inhibit their growth and flowering in order to endure the stress and continue development. In this study, the focus will be placed on the later strategy of stress tolerance and inhibition of flowering time as a response to water deficit.

The initiation of reproduction in plants happens on the basis of large amounts of internal and external cues [1,2,3,4]. Two types of strategies are recognized in *Arabidopsis* in relation to flowering time under drought stress—avoidance and tolerance. Lovell et al. [5] provided evidence that natural variation in the expression of *FRI (FRIGIDA)* correlates well with physiological parameters related to the two strategies. A low expression allele confers a “drought escape” strategy with fast growth, low water use efficiency, and early flowering, while “dehydration avoidance” is related to slow growth, efficient water use, and late flowering [5]. Utilizing a QTL population between Landsberg *erect* and Antwerp-1 [6] confirmed that early flowering as a drought escape strategy is related to strongly impaired plant fitness, while late-flowering plants were able to recover their growth in the second half of their vegetative development. Indeed, stress tolerance in the face of late flowering may be advantageous under conditions of mild water deficit [6]. Col-0, the wild type used as a control in the present study, is characterized as a “drought escape” phenotype with impairment in the *FRI* allele [5]. As such, it provides a good background for studying additional pathways that act to delay flowering time, which may be masked behind a strong *FRI* allele.

*FRI* is known to regulate flowering by promoting the expression of *FLOWERING LOCUS C* (*FLC*), via the establishment of active chromatin states [7]. *FLC* can then suppress the expression of *FLOWERING T (FT)* and *SUPPRESSOR OF OVEREXPRESSION OF CO 1 (SOC1)*. While Col-0 contains a weak *FRI* allele, there are other pathways that converge on the *FLC* locus. For example, some evidence exists that the *miRNA169*-regulated *NF-YA2* can promote the expression of *FLC* [8]. On the other hand, *NF-YA2* was also shown as a positive regulator of *FT* [9,10]. It is odd that the same transcription factor that positively affects the main gene responsible for the induction of reproductive growth also positively affects its main suppressor. Recent reviews also appear to have conflicting opinions regarding the effect of *NF-YA2* on flowering. Siriwardana [11] maintains the positive effects of the *NF-Y* complex on flowering, while Chen et al. [12] describe a model where *NF-YA2* promotes the expression of *FLC*, which in turn suppresses *FT*.

In another aspect of *FT* regulation, the B-box domain transcription factor *CONSTANS (CO)* is recognized as the main trigger of *FT* expression under long-day conditions [13]. Recently, two members of the B-box family, *CONSTANS-LIKE 7 (BBOX16/COL7)* and *CONSTANS-LIKE 8 (BBOX17/COL8)*, were found to interact directly with *CO* and inhibit its function as an *FT* promoter under long-day conditions [14,15]. The present study provides evidence that *COL7* and *COL8* become overexpressed under water deficit conditions, in a cytosine methylation-dependent manner, and may be responsible for the delay of flowering time observed in Col-0.

For the purpose of this study, the *drm1 drm2 cmt3* (*ddc*) triple methylation mutant in the Col-0 background was used. The DOMAINS REARANGED METHYLASE 1/2 (DRM1/2) are recognized as the main methyltransferases participating in the RNA-directed DNA methylation pathway in *Arabidopsis* [16,17]. In this context, the two enzymes are primarily responsible for cytosine methylation in euchromatic regions. On the other hand, CHROMOMETHYLASE 3 (CMT3) acts together with proteins from the SUVH family to form a self-reinforcing feedback loop with the histone modification H3K9me [18]. In this action, CMT3 is mainly responsible for cytosine methylation in CHG and CHH contexts, where H is any nucleotide except G. Additionally, *Arabidopsis* has the MET1 methyltransferase, which facilitates CG methylation maintenance in interaction with VIM1 and H3K9me2/3 [19], as well as the *CMT2* methyltransferase, which together with CMT3 facilitates CHG/CHH methylation [18], and together with DECREASED DNA METHYLAITON 1 (DDM1) facilitates methylation in heterochromatic regions [20]. Altogether, the *ddc* tripple knockout mutant is characterized by an almost complete loss of CHG methylation, a severe reduction in CHH, and a mild reduction in CG methylation [21]. Additionally, *ddc* was previously reported to exhibit a slight delay in flowering time under long-day conditions [22].

This study was performed under the hypothesis that if certain cytosine methylation patterns, established by the DRM1/2 and CMT3 methyltransferases and lost during water deficit, play an important role in the regulation of flowering time under water deficit in Col-0, the *ddc* mutant should not exhibit additional delays. Furthermore, differentially regulated genes responsible for the regulation of flowering time under water deficit should already be differentially expressed in the *ddc* mutant under control conditions and should share the same patterns of expression. If differential methylation is the main prerequisite for differential regulation under water deficit, then no differential expression is expected in the *ddc* mutant under stress. Here we show that, indeed, the COL7 and COL8 b-box domain proteins meet these criteria, together with the NF-YA2 transcription factor. Surprisingly, however, we did not find any changes in the cytosine methylation landscape within, or around the gene bodies, which could explain the changes in expression, indicating a *trans* regulatory mechanism. Considering that the activities of *COL7* and *COL8* in different physiological conditions are not elucidated, this paper provides a background for future experiments targeting the role of these genes in the fine-tuning of flowering time in *Arabidopsis thaliana*.

## 2. Results

### 2.1. Impact of Water Deficit on Flowering Time

To evaluate the interaction between cytosine methylation and water deficit on flowering time in Arabidopsis thaliana, a total of 160 plants were grown in a randomized complete block design. The control plants were watered twice a week, while drought was applied as a lack of water for two weeks, starting 12 days after seeding (DAS). A total of 96 plants were sampled 24 DAS for examination. At this stage, water deficiency had reduced the rosette biomass of Col-0 by approx. 60%, while the reduction in *ddc* was closer to approx. 50% (Figure 1A). While *ddc* naturally produces less biomass than Col-0, water deficit reduced the difference between the two genotypes by half. Very similar pattern was observed for rosette diameter (Figure 1B), with the significant difference of *ddc* being closer to Col-0 in diameter than in weight. This led us to hypothesize that *ddc* loses less water under drought than Col-0, leading to less weight loss and less water stress. To test this hypothesis, relative water content (RWC) was measured from leaf number 5 at 24 DAS from 85 plants. Surprisingly, we found that there is no significant difference between Col-0 and *ddc* in both the control and low-watering treatments (Figure 1C). Leaf number 5 was an already relatively old leaf at 24 DAS, and in some cases, it exhibited symptoms of senescence. This led to a relatively high biological variation, as visible on the graph (Figure 1C). Nevertheless, the visual assessment of the plants also indicated no wilting of the leaves, despite the significant reduction in plant growth. Significant reduction in plant growth was also combined with delayed bolting in Col-0, with an average of 1 day (Figure 1D). No impact on flowering time was observed in *ddc*, which was already exhibiting a delay between 1 and 2 days compared to Col-0 under control conditions. This indicates that the pathways responsible for delaying bolting in Col-0 may already be active in *ddc*. It is interesting to note that in both *ddc* and Col-0, the water deficit led to a more uniform bolting time for all plants, significantly reducing the number of both early and late flowering individuals.

### 2.2. Changes in Gene Expression Under Drought Stress

RNAseq was performed in order to test the hypothesis that a commonly up-regulated pathway between *ddc* under control conditions and Col-0 under low-watering treatment is responsible for the delay in bolting. The principal component analysis shows independent separation between the four treatment groups (Figure 2A), with PC1 separating the control from the low-watering treatment and PC2 separating Col-0 from ddc genotypes. We found 5239 differentially expressed genes in Col-0 under water deficit (Figure 2B; Appendix A) compared to 3307 in *ddc* (Figure 2D; Appendix A). This indicates that ddc might be experiencing less severe stress, as shown by the smaller growth reduction (Figure 1A,B). On the other hand, some pathways which are differentially regulated in *ddc* due to the mutation background may be influenced by drought stress in Col-0. In fact, there are 1356 differentially expressed genes in *ddc* compared to Col-0 under control conditions, with the majority being up-regulated genes (Figure 2C; Appendix A). All together, *ddc* show a different expression profile than Col-0, with 1166 up regulated and 784 down regulated genes when the two drought treatments are compared (Appendix A).

Next, we wanted to find genes whose patterns of expression match the patterns of bolting observed in Figure 1D. First, differentially expressed genes in Col-0 under the low-watering treatment compared to the control treatment were intersected with *ddc* compared to Col-0 under control. From these, only genes that are commonly up- or down-regulated were chosen, producing a set of 253 genes (Figure 3A,B). Finally, all genes that are differentially expressed in *ddc* under the low-watering treatment compared to control were removed, leaving 194 genes. No significant enrichment was found regarding biological processes when searching The Gene Ontology Resource [23,24]. Looking at molecular function, significant overrepresentation was found for glycosilase activity, with 10 differentially regulated genes related to the process. Looking at cellular components, 69 were chloroplast-related, with 73 related to plastids overall. Only 3 of the 194 genes were previously recognized regulators of flowering. These encode the B-box type zinc finger proteins CONSTANS-LIKE 7 (COL7/BBX16), CONSTANS-LIKE 8 (COL8/BBX17) and the NF-YA2 transcription factor (Figure 3C). Another gene was found, which showed suppressed gene expression, but was also significantly down-regulated in *ddc* under the low-watering treatment. This gene encodes the DELLA protein RGA-LIKE1 (RGL1) (Appendix A). Gene expression patterns for *COL7/BBX16*, *COL8/BBX17,* and *NF-YA2* were confirmed with RT-qPCR (Appendix A).

To find an alternative explanation for the changes in flowering time, we scanned the three pairwise comparisons of interest for differentially expressed genes related to flowering (Appendix A). In *ddc* compared to Col-0 under Control, *CCA1* was down-regulated, which also corresponded to induced expression of *TOC1*. Surprisingly, *CCA1* was up-regulated in both Col-0 and *ddc* under the low-watering treatment, indicating that down-regulation of *CCA1* is a methylation-specific response, which is reversed by water deficit. This, however, did not appear as a respective down-regulation of *TOC1*. Additionally, low watering enhanced the expression of *AP1*, *SNZ,* and *Erecta* and suppressed the expression of *AP2* in both Col-0 and *ddc*. The shared expression patterns indicate that these genes are likely not participating in the interaction between *ddc* and low watering regarding the induction of flowering. *GA* and *GAI* were significantly down-regulated in Col-0 under low-watering treatment compared to the control, which corresponded to the up-regulation of GA in *ddc* compared to Col-0 under the low-watering treatment. Overall, 19 genes were differentially expressed in Col-0 under low watering, compared to 9 genes in *ddc*. Only seven genes were differentially expressed in *ddc* compared to Col-0 under Control. These results underline the significant role DDC-induced cytosine methylation plays in regulating flowering time under water deficit.

### 2.3. Changes in Cytosine Methylation Under Water Deficit

To establish the role of cytosine methylation in the regulation of genes involved in bolting, whole genome bisulfite sequencing was utilized. The principal component analysis (PCA) shows little effect of the low-watering treatment on the cytosine methylation landscape (Figure 4A). The two large differences are between Col-0 and *ddc*, separated by PC1. PC2 shows the separation of the data points mainly based on the treatment differences and internal variation between the samples. While Col-0 samples become somewhat separated, *ddc* shows significant overlap between the two groups (Figure 4A). The statistical evaluation shows that there is an overall increase in methylation in all three cytosine contexts (Figure 4B–D). CG methylation shows an increase, which is non-significant in Col-0 and close to significance in *ddc* (Figure 4B). Both CHG and CHH methylation show significant increases in methylation levels following low watering (Figure 4C,D). These results indicate that de novo methylation as a response to the applied stress in the CG and CHH contexts is mostly DDC-independent. On the other hand, CHG de novo methylation is highly reduced, but not lost, in the *ddc* mutant. The mean CHG methylation for *ddc* is approx. 1.2% under low-watering conditions and 1.1% under control conditions. The mean CHG methylation for Col-0 is approx. 8.5% under low-watering and 8% under control conditions. This indicates that approx. 20% of the CHG *de novo* methylation in this experiment is DDC-independent.

Next, the cytosine methylation levels were plotted for *COL7* and *COL8*, within 1000 base pairs from the gene loci, including the 3′UTR and the 5′UTR, according to the TAIR10 assembly (Figure 5). *COL7* shows the existence of cytosine methylation in all three contexts, while *COL8* is methylated mostly at CHG and CHH. No differentially methylated loci or regions were found related to the two genes that could explain the observed changes in expression patterns (Appendix A). It is also interesting to note that *ddc* triple knockout leads to little, if any, change in the methylation landscape, with the exception of a significant increase in CG methylation towards the 3′UTR of *COL8*. These results indicate that methylation in these two genes is established and maintained in a manner independent of the DDC methyltransferases. Furthermore, the increase in gene expression resulting from the *ddc* mutations is likely not a direct cause of changes in methylation in *cis*, but rather a result of siRNA/miRNA production or changes in certain signal transduction pathways that impact the two genes in *trans*. Thus, this experiment shows that cytosine methylation plays a role in fine-tuning flowering time under water deficiency, potentially through regulating the expression of the COL7 and COL8 zinc finger proteins in *trans*. Further research is required to establish the exact mechanism by which *COL7* and *COL8* are regulated and the significance of their impact on flowering time under different stressful environments.

While not much is known about the regulation of *COL7* and *COL8* and their role in stress-induced changes in flowering time, *NF-YA2* is a relatively well-characterized gene. Under drought stress, *NF-YA2* is down-regulated via *miR169d* [25]. First, we mapped the methylation profile of the *NF-YA2* gene together with 1000 base pairs up- and downstream of the gene (Figure 6). No obvious changes in the methylation profile were found that could explain the expression under water deficit or in the *ddc* mutant. The gene appears to have CHG and CHH methylation levels comparable to those of *COL7* and *COL8* (Figure 5), again without much impact of the ddc mutations. Surprisingly, we found two CG methylation peaks towards the 3′UTR end of the gene and downstream of the *miR169d* target region. This region gained additional CG methylation in the *ddc* mutant (Appendix A). Within the target area of *mir169d,* however, significant gain of CG methylation was observed, but only for the ddc mutant and irrespective of the treatment (Figure 6, Appendix A). A slight reduction in CHG was also observed under water deficit in Col-0 and complete loss in *ddc*, but this was not deemed significant by the statistical model. Additionally, an increase in CHH methylation was observed downstream of the *miR169d* target region, which appeared in Col-0 under low-watering conditions and *ddc* under control conditions, but was lost in *ddc* under low-watering conditions. Whether these effects are stochastic in nature remains to be elucidated. The gene expression patterns and the phenotype in the present experiment suggest that these changes in cytosine methylation are not a cause but rather an effect of the interaction between miR169d and the *ddc* triple knockout background.

## 3. Discussion

There are two general strategies *Arabidopsis thaliana* plants utilize in order to cope with drought stress—drought escape and drought tolerance. The wild-type Col-0, which is often used in laboratory experiments, is a drought escape accession with a nonfunctional *FRI* allele resulting in earlier flowering [5]. We observed that, in its growth pattern under water deficiency, Col-0 does not lose turgor for a long time (Figure 1C) [26]. Rather, it slows down growth and delays further development (Figure 1A,B). Transition to flowering still occurs but with a delay (Figure 1D). When water deficit becomes critical, the plants lose turgor rapidly and die.

We have previously observed that the *ddc* (*drm1/drm1/cmt3*) triple knockout mutant shows a slight delay in its transition to flowering under long-day conditions [22]. In the present study, we tested if the delay from the *ddc* mutations is additive to that from water deficit and found that it is not (Figure 1D). The first obvious gene candidate for this effect is the *FLC*. It has been long known that *FLC* is under *trans-*acting control from DNA methylation, but the exact mechanism remains to be elucidated [27,28]. Nevertheless, hypomethylation is generally related to *FLC* down-regulation [27]. On the other hand, drought stress was shown to induce *FLC* expression in an ABA dependent manner [29]. Indeed, we found that *FLC* is significantly up-regulated in the Col-0 wild type under water deficit, but not under the *ddc* control, or *ddc* low-watering conditions (Appendix A). It is important to note that there is no significant difference between *FLC* expression in Col-0 low watering and *ddc* low watering, indicating small increases in expression in *ddc* compared to Col-0 under control conditions, and *ddc* low watering compared to the control, which are not considered significant by the statistical model.

In *ddc* compared to Col-0 under control conditions, we found a slight but significant overexpression of *SOC1* (Appendix A). This gene is long known as a central regulatory hub for the induction of flowering in *Arabidopsis,* together with *FT* [30]. Another surprising finding in this comparison is the up-regulation of *TOC1* and the corresponding down-regulation of *CCA1* (Appendix A). *TOC1* is generally recognized as a stabilizer of *CO* and promoter of flowering, while *CCA1* itself is a negative regulator of *FT* and *SOC1* [31,32,33,34]. Additionally *CCA1* appears to be up-regulated in both *ddc* and Col-0 under water deficit, indicating that cytosine methylation-dependent and stress-related gene regulation act via two distinct pathways. *CCA1* up-regulation could in fact correspond to delayed flowering time under low-watering conditions; however, the lack of additional effect in *ddc* indicates another pathway which overrules the effects of *CCA1*.

Another part of the regulation of flowering time recognized in this study is the *NF-YA2* gene, part of the NF-Y conserved transcriptional regulator complex. Previous reports suggest that *NF-YA2* promotes *FLC* expression [8,12] and therefore suppresses the transition to reproductive growth. In our experiment *NF-YA2* expression is down-regulated in Col-0 under water deficit and in *ddc* under control and water deficit conditions. However, this does not match the expression patterns of *FLC*. On the other hand, evidence exists that *NF-YA2* promotes expression of *FT* via direct interaction with the *FT* promoter [9]. *FT* expression in the present study is not differentially regulated in any of the conditions. This is most likely due to harvesting of the plant material several hours after the beginning of their day, resulting in low base reads from the *FT* gene (Appendix A).

Recent studies brought insight into the control of *NF-YA2* under drought and heat stress [25]. The researchers demonstrated that overexpression of *miR169d* under drought stress resulted in the down-regulation of the *NF-YA2* gene. We found that down-regulation of the *NF-YA2* gene is additionally correlated with some interesting changes in the cytosine methylation landscape, within and downstream of the location targeted by *miR169* (Figure 6, Appendix A). The appearance of CG methylation peaks in methylation deficient mutants, including *ddc,* was described as early as 2005 [35]. Here we observe that such a peak can appear as a result of the action of miRNA silencing in the *ddc* background. The mechanisms behind this remain to be elucidated.

Finally, two b-box type zinc finger proteins were found to be overexpressed in Col-0 under water deficit and in *ddc* under control conditions. These are the *BBX16* and *BBX17,* otherwise known as *CONSTANS-LIKE 7 (COL7)* and *CONSTANS-LIKE 8* (COL8), respectively. Recently, BBX16/COL7 was recognized as a regulator of flowering time via direct interaction with the CO protein in the nucleus [14]. The BBX-CO interaction significantly reduced the ability of CO to induce *FT* expression and was therefore recognized as a negative regulator of flowering time. Additionally, the overexpression of *BBX16/COL7* resulted in delayed flowering in *Arabidopsis*. One year earlier it was shown that BBX17/COL8 has a similar function of interacting with the CO protein and inhibiting its activity [15]. To our knowledge, there is no data showing the involvement of these two genes in flowering regulation under water deficit.

The cytosine methylation data does not show any obvious alterations that could cause the differential expression patterns observed in this study. The significant increase in CG methylation towards the 3′ end of *COL8 however,* resembles the increase observed in *NF-YA2*. Additionally, there is an inconsistency between TAIR10 (used in this study) and the Araport11 assembles. Araport11 indicates that the *COL8* transcript expands approx. 400 bp longer as a 3′UTR towards the second *ddc*-dependent CG methylation peak (Figure 5). This observation poses the question of the potential regulation of *COL8* via an unknown miRNA pathway. The proof of the CG hypermethylation peaks resulting from the activity of miRNAs in the *ddc* background, however, is outside of the scope of this manuscript.

## 4. Materials and Methods

### 4.1. Plant Growth

The seeds for this experiment were obtained from the Nottingham Arabidopsis Stock Center—NASC. The *ddc* triple knockout mutant (NASC ID: N16384) was grown for one generation for seed production and then utilized for this experiment. Approx. 5–10 seeds were placed in a 130 mL pot, filled with a moist mixture of 3 parts peat substrate to 1 part perlite. The seeds were stratified for 48 h in complete darkness at 4 °C. After that, the plants were grown under long days (16 h day/8 h night) at 22 °C and 70% relative humidity. After germination, the pots were thinned until 1 plant per pot was left. Col-0 and *ddc* plants were completely randomized within blocks. Each block consisted of 2 trays next to each other, one of them used as a control and the other as a low-watering treatment. The control was watered two times a week. For the treatment, watering was stopped for 12 days starting at day 12 after seeding. Due to the small size of the pots, the soil completely dried out during the 12 days without watering. The total number of plants used was 160. Twenty-four days after seeding (DAS), before watering, the whole leaf rosette of 96 plants was harvested for further analysis. At this stage all of the harvested plants had already transitioned to reproductive growth, but did not have their first flowers. Bolting and rosette diameter of all plants was estimated visually. Bolting was evaluated in DAS when the shoot apical meristem has visibly transitioned from vegetative to reproductive growth. Rosette diameter was evaluated from pictures using ImageJ 1.54g. After harvesting, the plants’ rosette fresh weight was estimated. Leaf number five was used for relative water content. The rest of the plant material was frozen in liquid nitrogen and stored at −80 °C.

### 4.2. Relative Water Content

Relative water content from leaf number five was estimated with the following formula: $\frac{FW-DW}{TW-DW}$. Here, *FW* is the leaf fresh weight; *DW* is the leaf dry weight; and *TW* is the leaf turgid weight after 24 h in distilled water, before the beginning of the dehydration process.

### 4.3. RNA Extraction

For RNA extraction, the 96 plants were pooled in 12 samples and ground in liquid nitrogen. There wertr four treatment conditions (Col-0 Control, Col-0 low watering, *ddc* Control, and *ddc* low watering) with three biological replicates. RNA was extracted with PRImeZOL™ from Canvax, Valladolid, Spain, according to the manufacturer’s instructions, with slight modifications. A total of 40–50 mg frozen plant material was placed in 1 mL of PRImeZOL™ and homogenized for 1 min in a VWR Star-Beater at a frequency of 30/s. The samples were centrifuged at 12,000× *g* for 10 min at 4 °C. The supernatant was transferred to a new tube and incubated for 1 min at room temperature. A total of 200 µL of chloroform was added; the sample was mixed by pipetting up and down carefully and incubated for 3 min at room temperature. After centrifugation at 12,000× *g* for 15 min at 4 °C, approx. 400 µL of the upper aqueous phase was moved to a new tube. An amount of 500 µL of isopropanol was added, and the samples were incubated for 10 min at room temperature. Following 10 min of centrifugation at 12,000× *g* and 4 °C, the supernatant was carefully removed and two steps of washing with 75% ethanol ensued. Washing was performed with 10 min incubation with shaking at 200 rpm and room temperature, followed by centrifugation at 7500× *g* for 5 min at 4 °C and a change of the supernatant. After that, the pellet was air-dried for approx. 15 min, or until it lost its whitish color and became transparent. The RNA was dissolved in 50 µL of RNAse-free water and incubated for 10 min at 60 °C.

### 4.4. RNAseq

The RNA from the 12 samples was sequenced by BGI Genomics on the DNBSEQ platform with PE150 sequencing read length and DNBSEQ Eukaryotic Strand-Specific mRNA library (Appendix A). The raw data was filtered with SOAPnuke v2.1.9 [36] with the following parameters: *-n 0.001-l 20-q 0.4—adaMR 0.25—polyX 50–minReadLen 150*. The read quality was checked using FastQC v0.11.9 (Andrews 2017) [37]. Genome indexing and alignment were performed with STAR v2.7.11b [38]. The following parameters were used for indexing:*—sjdbOverhang 299—genomeSAindexNbases 12*. The following parameters were used for alignment:*—outSAMtype BAM Unsorted—quantTranscriptomeBan Singleend—outFilterType BySJout—alignSJoverhangMin 8—outFilterMultimapNmax 20—alignSJDBoverhangMin 1—outFilterMismatchNmax 999—outFilterMismatchNoverReadLmax 0.04—alignIntronMin 20—alignIntronMax 6000—alignMatesGapMax 1000000—quantMode TranscriptomeSAM—outSAMattributes NH HI AS NM MD*. Quantification was then performed using Salmon v1.10.1 [39]. The TAIR10 genome assembly was used for this analysis. 

### 4.5. RT-qPCR

Gene expression for *COL7/BBX16*, *COL8/BBX17,* and *NF-YA2* was confirmed with the help of RT-qPCR. Briefly, RNA was extracted from the remaining pooled leaf samples as described earlier. cDNA was synthesized with RevertAid H Minus First Strand cDNA Synthesis Kit from thermo scientific, according to the manufacturer’s instructions. The RT-qPCR was performed using Sso Fast EvaGreen™ Supermix from Bio-Rad, Hercules, CA, USA, according to the manufacturer’s instructions. The data was normalized with ∆∆Ct normalization towards two reference genes and Col-0 under control conditions. The following primer pairs were used:

Tubulin

F 5′-CACATTGGTCAGGCCGGTAT-3′

R 5′-GCACCGGTCTCACTGAAGAA-3′

Actin

F 5′-AGAGATTCAGATGCCCAGAAGTCTTGTTCC-3′

R 5′-ACGATTCCTGGACCTGCCTCATCATACTC-3′

COL7/BBX16

F 5′-CGATGACGCTTTCCTATGCC-3′

R 5′-GTTTTGTCTGCCGTCTCCGT-3′

COL8/BBX17

F 5′-AAGATGTCAAGCAGCCACGA-3′

R 5′-TCGGAGGCACAGTACCAAAC-3′

NF-YA2

F 5′-GTGAACTCAAAGCAATACCATGG-3′

R 5′-TGGTTCCGCTATTTTCCAAGT-3′

### 4.6. DNA Extraction

DNA was extracted from the same 12 samples used for RNAseq. Approx. 100 mg of plant material were placed in 500 µL CTAB extraction buffer (2% cetyl trimethylammonium bromide, 1% polyvinylpyrrolidone, 100 mM Tris-HCl, 1.4 M NaCl, 20 mM EDTA), vortexed thoroughly and incubated at 60 °C for 30 min. After centrifugation for 5 min at 14,000× g, the supernatant was transferred to a new tube, and 5 µL of RNAse A solution was added. The samples were incubated for 20 min at 37 °C. After that, 500 µL of phenol–chloroform–isoamyl alcohol (25:24:1) was added and mixed by pipetting. The phases were separated via centrifugation for 1 min at 14,000× *g*. The upper aqueous phase was moved to a new tube, and 0.7 volumes of cold isopropanol was added. Precipitation was enhanced at −20 °C for 15 min, and DNA was pelleted at 14,000 g for 10 min. The isopropanol was removed, and two subsequent washing steps with 75% ethanol were performed, with 10 min incubation time for each. The DNA pellet was air-dried for approx. 15–20 min and dissolved in 50 µL TE buffer (10 mM Tris, pH 8, 1 mM EDTA).

### 4.7. Whole Genome Bisulfite Sequencing

The DNA from the 12 samples was sequenced by BGI Genomics on the DNBSEQ platform with PE150 sequencing read length and DNBSeq Whole genome bisulfite library (Appendix A). The raw data was filtered with SOAPnuke v2.1.9 [36] with the following parameters: *-n 0.001-l 20-q 0.4—adaMR 0.25—ada_trim–minReadLen 150*. The read quality was checked using FastQC v0.11.9 [37]. Sequence alignment was performed using Bismark v0.24.2 [40] with the following settings:*—multicore 8—score_min L,0,-0.4*. The alignments were then sorted with Samtools [41]. The data was deduplicated, and cytosine methylation information was extracted with Bismark. Methylation extraction was performed with the following settings:*-p—ignore 3—ignore_r2 3—comprehensive—bedGraph—CX—multicore 10*. The TAIR10 genome assembly was used for this analysis.

### 4.8. Statistical Analyses

All statistical analyses and data visualizations were performed on R 4.3.1. Rosette fresh weight, rosette diameter, relative water content, bolting and global methylation levels in CG, CHG, and CHH were evaluated using a mixed linear model. Treatment, genotype, and their interaction were used as main effects, while the block structure of the experiment was used as a random effect. This was performed with the help of the *lme4* package v1.1-36 [42]. Estimated marginal means and contrasts were calculated with the *emmeans* package v1.10.7 [43]. The data was visualized with the help of *ggplot2* v3.5.1 [44], *ggpubr* v0.6.0 [45], and *cowplot* v1.1.3 [46].

Statistical analysis of the RNAseq data was performed using the *DEseq2* v1.40.2 pipeline [47]. All genes with less than 10 reads in more than 9 samples were removed from the analysis. PCA was performed and visualized with the *plotPCA* function from the *DEseq2* package. Differential expression was calculated, accounting for the block effect of the experimental design. Volcano plots were made using *ggplot2* and *ggpubr*. The Venn diagram was visualized with the help of *ggVennDiagram* v1.5.2 [48].

Cytosine methylation levels were calculated as a proportion of methylated over total reads. PCA was performed on the whole genome bisulfite sequencing data using the *factoextra* package v1.0.7 [49]. For this, all methylation levels were averaged out on 100 bp bins by cytosine context. Raw methylation levels were plotted out for the regions encompassing 1000 bp upstream and downstream of the *COL7* (AT1G73870), *COL8* (AT1G49130) and *NF-YA2* (AT3G05690) genes, according to the TAIR10 assembly, using *ggplot2* and *ggpubr*. Statistical significance for differential methylation was estimated with a mixed linear model (*lme4*) evaluating treatment, genotype, and their interactions, as well as the block structure as a random effect. For the search of differentially methylated loci (DML), this was performed for every single cytosine. A rolling window approach was used for the location of differentially methylated regions (DMR), where the mean methylation for 100 bp was estimated and used for the statistical analysis on a per cytosine context basis.

## 5. Conclusions

The results in this experiment support a model of delayed flowering in Col-0 under water deficit by the inhibition of the *FT* inducers *CO* and *NF-YA2*. Under long-day conditions, the BBX16/COL7 and BBX17/COL8 proteins interact with CO to inhibit its function as an *FT* promoter (Figure 7). On the other hand, *miR169d* overexpression down-regulates *NF-YA2*, which is in turn an independent promoter of *FT*. Col-0 contains a non-functional *FRI* allele and is therefore unable to induce significant inhibition of the *FT* gene via *FLC*. This results in the suppression of the transition to reproductive growth which, however, can be easily overcome. The end effect is the slight delay in flowering time observed in this experiment. No additional delay in flowering time in *ddc*, together with matching patterns of expression of the three genes, indicate the significant role of cytosine methylation in this process. How exactly does cytosine methylation impact the expression of *BBX16/COL7, BBX17/COL8,* and *NF-YA2* in *trans*, remains to be elucidated.

## Figures and Tables

**Figure 1 ijms-26-08360-f001:**
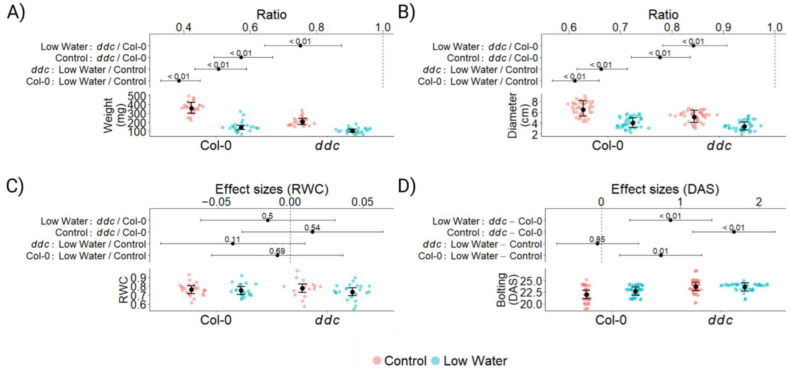
Plant growth and phenotype of *Arabidopsis thaliana* wild type Col-0 and *ddc* triple knockout mutant under drought stress. The data was analyzed using mixed linear models. Red dots represent the control treatment; blue dots represent the low-watering treatment. Black dots are the estimated marginal means, with whiskers representing the 95% CI. The numbers on the graph represent *p values* of the differences produced by the linear models. Logarithmic transformation was applied to the weight and rosette diameter for better model fit. Therefore, the resulting effect sizes are in the form of ratios. (**A**) Plant weight at 24 DAS, measured in mg; *n* = 96. (**B**) Rosette diameter at 24 DAS, measured in cm; *n* = 160. (**C**) Relative water content (RWC) at 24 DAS; *n* = 85. (**D**) Time of bolting measured in days after seeding (DAS); *n* = 160.

**Figure 2 ijms-26-08360-f002:**
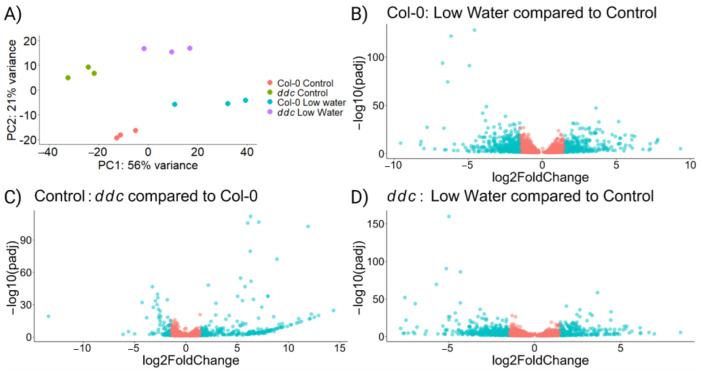
Gene expression of *Arabidopsis thaliana* wild type Col-0 and *ddc* triple knockout mutant under drought stress. A total of 48 plant samples were pooled in 4 groups with 3 biological replicates each. (**A**) PCA performed on the log transformed gene expression data. (**B**) Volcano plot for differentially expressed genes under low-watering treatment compared to control conditions in Col-0 wild type; *n* = 5239. (**C**) Volcano plot for differentially expressed genes in *ddc* compared to Col-0 under control conditions; *n* = 1356. (**D**) Volcano plot for differentially expressed genes under low-watering conditions compared to control conditions in *ddc* triple knockout mutant; *n* = 3307. The genes represented on the volcano plots were selected with *padj* < 0.05. Red-colored dots represent genes with log2FoldChange < 1.5; blue-colored dots represent genes with log2FoldChange ≥ 1.5.

**Figure 3 ijms-26-08360-f003:**
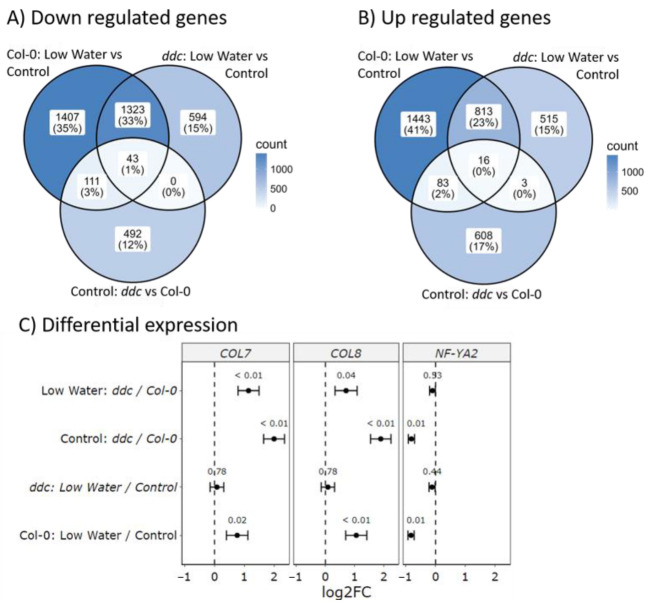
Gene expression analysis of Col-0 wild type and *ddc* triple knockout mutant under drought stress. Venn diagrams of (**A**) up-regulated and (**B**) down-regulated differentially expressed genes in three pairwise comparisons: Col-0: low-watering vs. control, *ddc*: low-watering vs. control, Control: *ddc* vs. Col-0. (**C**) Differential expression of *COL7*, *COL8,* and *NF-YA2*. Black dots represent the mean log2 fold change (log2FC), whiskers represent the standard errors of the log2FC, and the small numbers above the means represent the adjusted *p* values.

**Figure 4 ijms-26-08360-f004:**
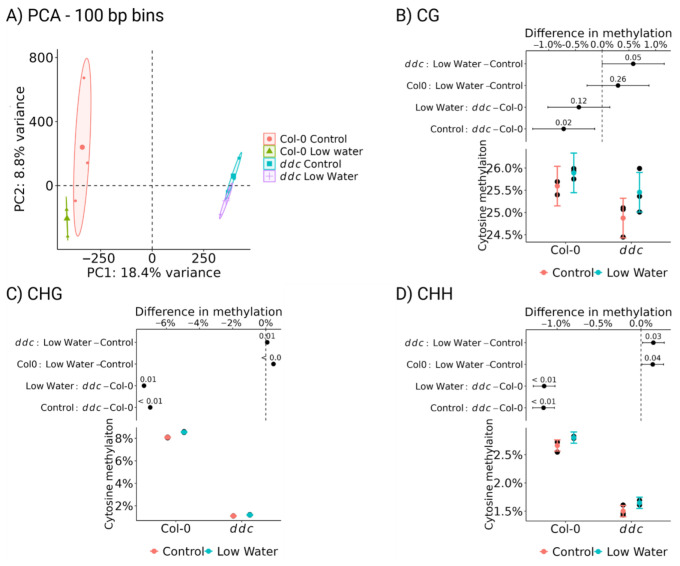
Whole genome bisulfite sequencing data of Col-0 wild type and *ddc* triple knockout mutant under drought stress. A total of 48 plant samples were pooled in 4 groups with 3 biological replicates each. (**A**) PCA on the methylation levels of all cytosine contexts averaged out using 100 bp bins. Mean cytosine methylation of the (**B**) CG, (**C**) CHG, and (**D**) CHH contexts is presented as estimated marginal means with 95% CI (bottom), along with the mean difference levels, 95% CI, and *p* value of the mean difference (top).

**Figure 5 ijms-26-08360-f005:**
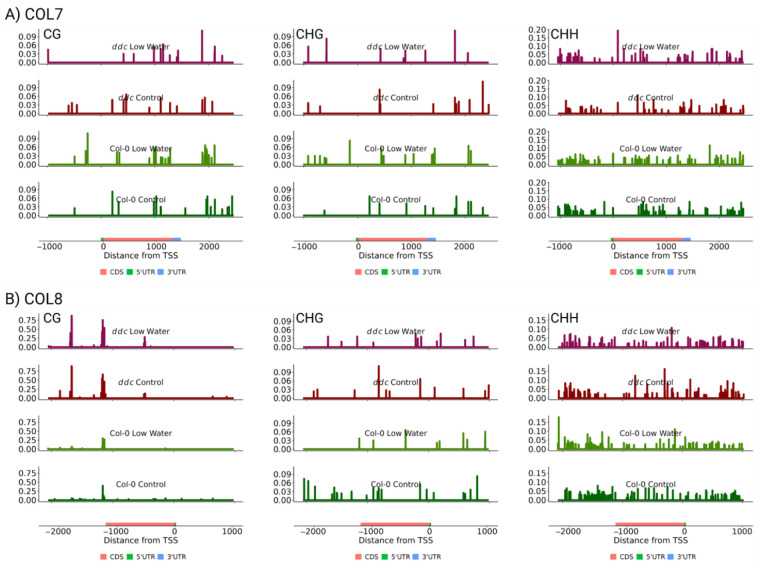
Mean cytosine methylation levels from three biological replicates of CG (left), CHG (middle), and CHH (right) contexts for (**A**) *COL7* and (**B**) *COL8*. On the x axis, transcription start site is labeled as 0, with position numbers increasing according to the positive DNA strand. Y axis measures cytosine methylation proportion for each individual cytosine. The four experimental conditions are presented from top to bottom as follows: *ddc* low watering, *ddc* control, Col-0 low watering and Col-0 control.

**Figure 6 ijms-26-08360-f006:**
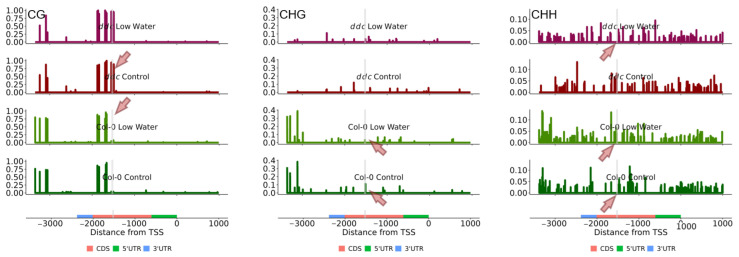
Mean cytosine methylation levels from three biological replicates of CG (left), CHG (middle), and CHH (right) contexts for *NF-YA2*. The four experimental conditions are presented from top to bottom as follows: *ddc* low watering, *ddc* control, Col-0 low watering and Col-0 control. Red arrows point at the location of *miR169d* interaction, where changes in the cytosine methylation profile are observed. Light gray lines indicate the target site of *miR169d*.

**Figure 7 ijms-26-08360-f007:**
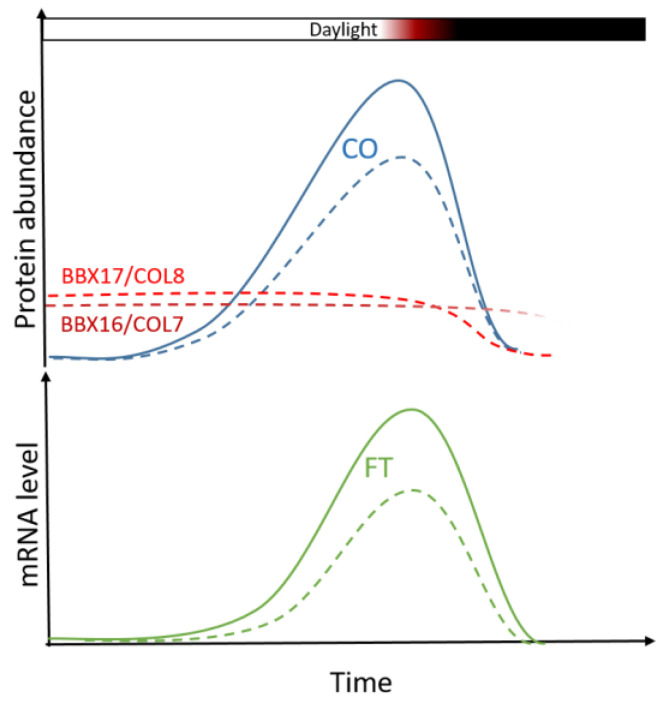
Under normal conditions, the expression of *CO* peaks during the late hours of the day, lead to the accumulation of the CO protein. During the night, the expression of *CO* is inhibited, while its protein product becomes actively degraded. Long days lead to induction of flowering due to the accumulation of *CO* before sunset (solid blue line), which in turn induces the expression of the florigen *FLOWERING T* (*FT*; solid green line). Drought stress induces the expression of the *BBX16/COL7* (dashed dark red line) and *BBX17*/*COL8* (dashed red line) genes, whose products interact with CO and inhibit its function as an *FT* inducer (dashed blue line), leading to reduction in FT expression (dashed green line) and delays in bolting. It is known that BBX17/COL8 becomes actively degraded during the night in a *COP1*-dependent manner. The daily patterns of expression and product accumulation of *BBX16/COL7* and *BBX17/COL8* remain to be elucidated.

## Data Availability

The original data presented in the study are openly available in SRA of NCBI with accession number PRJNA1301877. Additional data supporting the conclusions of this article can be made available by the authors on request.

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
