# Peer review of "Delay in Flowering Time in *Arabidopsis thaliana* Col-0 Under Water Deficit and in the *ddc* Triple Methylation Knockout Mutant Is Correlated with Shared Overexpression of *BBX16* and *BBX17"

_ijms, 2025, doi:10.3390/ijms26178360_

Round 1
Reviewer 1 Report
Comments and Suggestions for Authors
The authors integrated transcriptomic analysis and methylation sequencing to investigate how drought stress affects the flowering time of Arabidopsis thaliana through epigenetic mechanisms.
The authors need to upload the raw sequencing data to a public repository (e.g., NCBI SRA).
The authors state: "For the treatment, watering was stopped for 12 days starting at day 12 after seeding." However, the drought intensity (e.g., soil moisture percentage) and duration are not specified, potentially compromising the reproducibility of the results. The current data represent only a single time point after drought treatment compared to a control, which may lead to coincidental findings. Furthermore, **it is crucial that the authors specify: At what time point was sampling performed, and which tissue(s) were used for transcriptome and methylation sequencing?** Because even if plants are the same chronological age, developmental stages and growth states may not be synchronized across individual plants or plant parts, potentially leading to different genes being affected.
The current conclusions rely primarily on correlations between gene expression and methylation levels, lacking direct evidence of molecular interactions. Validating these findings using mutants such as *col7*, *col8*, and *nf-ya2* would be more convincing.
The comparisons between different groups in Figure 2 are unclear.And the statistical analysis would be more appropriate using a two-way ANOVA method.
In the subsequent analysis, only three pairwise comparisons were made for gene expression differences: Col-0: Low Water vs Control, ddc: Low Water vs Control, and Control: ddc vs Col-0. However, a significant phenotypic difference was also observed for the comparison Low Water: ddc vs Col-0.
Lines 154-156: The authors state that 253 common Differentially Expressed Genes (DEGs) were identified by three pairwise comparisons: Col-0: Low Water vs Control, ddc: Low Water vs Control, Control: ddc vs Col-0. Subsequently, genes that were differentially expressed in ddc under low water conditions compared to its control were excluded, resulting in 188 genes. The corresponding data representation for this filtering step (253 -> 188) or the final set of 188 genes is not clearly presented in Figure 4A.
Author Response
Comments 1: The authors need to upload the raw sequencing data to a public repository (e.g., NCBI SRA).
Response 1: We have uploaded the RNA-seq dataset in the NCBI SRA, accession No. PRJNA1301877
Comments 2: The authors state: "For the treatment, watering was stopped for 12 days starting at day 12 after seeding." However, the drought intensity (e.g., soil moisture percentage) and duration are not specified, potentially compromising the reproducibility of the results. The current data represent only a single time point after drought treatment compared to a control, which may lead to coincidental findings. Furthermore, **it is crucial that the authors specify: At what time point was sampling performed, and which tissue(s) were used for transcriptome and methylation sequencing?** Because even if plants are the same chronological age, developmental stages and growth states may not be synchronized across individual plants or plant parts, potentially leading to different genes being affected.
Response 2: We added a statement that clarifies the usage of small pots for growth of individual plants:
‘Approx. 5-10 seeds were placed in a 130 ml pot, filled with a moist mixture of 3 parts peat substrate to 1 part perlite’
Afterwards we added the following statement specifying that the pots dry out completely during the 12 days period without watering:
‘Due to the small size of the pots, the soil completely dried out during the 12 days without watering’
Furthermore we state that the whole leaf rosette was harvested for plants that have already transitioned to reproductive growth. Specifically plants that are during the stage of bolting, before the opening of the first flowers.
‘At 24 days after seeding (DAS), before watering, the whole leaf rosette of 96 plants was harvested for further analysis. At this stage all of the harvested plants had already transitioned to reproductive growth, but did not have their first flowers.’
We have repeated this experiment multiple times and have found this arrangement to be reliably repeatable in our conditions. Due to the fact that the soil dries out completely in the 12 day period without watering, we decided to measure leaf relative water content (RWC) instead. The experimental setup was designed so that the plants do not dry out and die before transition to reproductive growth. This is indicated by the results of the RWC, where leaf water content remains stable, regardless of the significant reduction in rosette growth.
Comments 3: The current conclusions rely primarily on correlations between gene expression and methylation levels, lacking direct evidence of molecular interactions. Validating these findings using mutants such as *col7*, *col8*, and *nf-ya2* would be more convincing.
Response 3: We completely agree with the reviewer and appreciate the lack in our current manuscript. This is the reason we have been careful not to make concrete conclusions, but specify in the title that the study is about correlation. We are currently planning on continuing the research and are waiting for knockout mutants of the mentioned genes. It is possible, that if 2, or more of these genes are responsible for the observed effect, a double mutant and an overexpression line of NF-YA2 will have to be generated in order for a strong phenotype to occur. This is why we are planning a bigger scale experiment for the future, where we will investigate in depth the roles of the three genes in the regulation of flowering time under drought stress.
Comments 4: The comparisons between different groups in Figure 2 are unclear.And the statistical analysis would be more appropriate using a two-way ANOVA method.
Response 4: In the present data analysis we have chosen to use a linear model instead of ANOVA due to the ability to draw more comprehensive conclusions from the results. In the following figure we represent the mixed linear model which was used (Figure 1A) compared to a two-way ANOVA (Figure 1B). While the two-way ANOVA uses F-statistics, the linear model uses t-statistics. This leads to small differences in the estimated p-values. Nevertheless, the two methods show statistical significance for Treatment, Genotype and their interaction. The linear model has the advantage of showing the effect size and its standard error in the fields Estimate and Std. Error (Figure 1 A). From these two fields we can calculate the difference between e.g. Col-0 Control and ddc Control and we can give an estimate on how sure we are about it. The 95% CI represented on Figure 2 (now Figure 1) of the manuscript represent our confidence in the size of the effect, considering the results of the experiment. The reason why Figures 2A and B (now Figures 1A and B) have ratio and Figures 2C and D (Now Figures 1C and D) have effect sizes is due to the log normalization of the data for weight and rosette diameter. . A comparable approach would be to perform e.g. TukeyHSD on the two-way ANOVA and present the results. However, this also does not provide an estimate of the effect size, but only the statistical significance of the means and their differences. Using the present approach we can say with confidence that e.g. low watering led to a significant reduction in rosette weight of Col-0 (p < 0.01) with between approx. 65% and 55%. We can provide the results of the linear models used in the form presented here (Figure 1).
Comments 5: In the subsequent analysis, only three pairwise comparisons were made for gene expression differences: Col-0: Low Water vs Control, ddc: Low Water vs Control, and Control: ddc vs Col-0. However, a significant phenotypic difference was also observed for the comparison Low Water: ddc vs Col-0.
Response 5: We have shown the three comparisons, because mainly they were used for finding genes that can explain the observed phenotype. To account for the missing representation, we have changed Figure 4 B) to Figure 3 C), which now includes the fourth comparison mentioned by the reviewer. Furthermore, we have added a supplementary table outlining the differential expression of ddc compared to Col-0 under Low Watering treatment and have included it in the text (Supplementary Table 6).
Comments 6: Lines 154-156: The authors state that 253 common Differentially Expressed Genes (DEGs) were identified by three pairwise comparisons: Col-0: Low Water vs Control, ddc: Low Water vs Control, Control: ddc vs Col-0. Subsequently, genes that were differentially expressed in ddc under low water conditions compared to its control were excluded, resulting in 188 genes. The corresponding data representation for this filtering step (253 -> 188) or the final set of 188 genes is not clearly presented in Figure 4A.
Response 6: To mitigate the issue we have now split Figure 4A into Figures 3A and B, for common up- and down- regulated genes. During the repeated analysis we identified a small error in the underlying script, which excludes several additional genes, to make the total from 188 to 194. This resulted in significant overrepresentation of glycosilase activity when probing The Gene Ontology Resource for molecular function. The text was altered correspondingly.

Reviewer 2 Report
Comments and Suggestions for Authors
Dear authors,
your manuscript "Delay in flowering time in Arabidopsis thaliana Col-0 under water deficit and in the ddc triple methylation knockout mutant is correlated with shared overexpression of BBX16 and BBX17" represents an interesting fundamental study that deepens our understanding of the genes involved in the regulation of flowering.
The study is certainly expands our knowledge of the regulation of generative development in plants.
The obtained material is competently discussed. The conclusion corresponds to the data obtained and postulates that delayed flowering under water deficit may be associated with increased expression of BBX16/COL7 and BBX17/COL8, the negative regulators of CO gene . This, in turn, causes a consistent change in the expression of other genes involved in regulating the timing of flowering and the transition of the apical meristem from vegetative to generative, including florigen - FT gene.
It would also be interesting to discuss the changes in the expression of other genes that you discovered, for example, GI, GAI, TOC1, CCA1, COP1, AP1, AP2, LFY. All of them are involved in the induction of flowering and the formation of floral meristem.
Some minor comments:
1) in Fig. 1 it is not clear where which genotypes are;
2) according to the photo (Fig. 1), plants with water deficiency retain their green color better than with normal watering. How can this be explained? You noted that among the DEGs there are genes of chloroplasts and other plastids. Is this related to the delayed flowering and aging in the experimental plants?
I would like to point out again that your research is interesting and significant.
Author Response
Comments 1: It would also be interesting to discuss the changes in the expression of other genes that you discovered, for example, GI, GAI, TOC1, CCA1, COP1, AP1, AP2, LFY. All of them are involved in the induction of flowering and the formation of floral meristem.
Response 1: We have included analysis for the TOC1, CCA1, AP1 and AP2 in the result section of the text, with TOC1 and CCA1 participating in the discussion section as well. COP1 and LFY are not differentially expressed in any of the conditions in our dataset and therefore we have not mentioned them. We recognize that we have failed to mention GA and GAI. We included the following statement in the results section:
‘GA and GAI were significantly down regulated in Col-0 under low watering, compared to control, which corresponded to and up regulation of GA in ddc compared to Col-0 under Low Watering.’
The shared response of AP1, AP2, SNZ and Erecta means that these genes are likely not participating in the interaction between ddc and Low Watering regarding the induction of flowering. We have added this clarification in the results section as follows:
‘Additionally, Low Watering enhanced the expression of AP1, SNZ and Erecta and suppressed the expression of AP2 in both Col-0 and ddc. The shared expression patterns indicate that these genes are likely not participating in the interaction between ddc and Low Watering regarding induction of flowering.’
Comments 2: 1) in Fig. 1 it is not clear where which genotypes are;
Response 2: We agree with the reviewer that Figure 1 is unclear, therefore we removed it.
Comments 3: 2) according to the photo (Fig. 1), plants with water deficiency retain their green color better than with normal watering. How can this be explained? You noted that among the DEGs there are genes of chloroplasts and other plastids. Is this related to the delayed flowering and aging in the experimental plants?
Response 3: To answer the reviewer’s question we looked at 44 genes related to photosynthesis and found that 8 of them are down regulated in Col-0 under Low Water compared to Control. Only one of them is down regulated in ddc Low Water compared to Control. Additionally, we have 10 up regulated and 4 down regulated genes in ddc compared to Col-0 under Control. Under Low Watering we have 19 up regulated genes in ddc compared to Col-0. This gene expression profile can explain the elevated chlorophyll concentration we have observed for the mutant compared to the wild type in various studies here at the institute, but correspond to reduction in photosynthetic capacity of Col-0 under Low Water compared to Control. Therefore, we can speculate that the darker coloration is caused by limited cell expansion and leaf growth due to the stress conditions resulting in higher concentration, despite reduction in photosynthetic capacity. How this may affect flowering time is unclear.

Round 2
Reviewer 1 Report
Comments and Suggestions for Authors
That's OK.